# Synthesis and Properties of Hydrophilic and Hydrophobic Deep Eutectic Solvents via Heating-Stirring and Ultrasound

**DOI:** 10.3390/molecules29133089

**Published:** 2024-06-28

**Authors:** María Isabel Martín, Irene García-Díaz, María Lourdes Rodríguez, María Concepción Gutiérrez, Francisco del Monte, Félix A. López

**Affiliations:** 1Centro Nacional de Investigaciones Metalúrgicas (CENIM-CSIC), Avda. Gregorio del Amo, 8, 28040 Madrid, Spain; irenegd@cenim.csic.es (I.G.-D.); lourdesrodriguezcea@cenim.csic.es (M.L.R.); 2Instituto de Ciencia de Materiales de Madrid, C. Sor Juana de la Cruz, 3, Campus de Cantoblanco, 28049 Madrid, Spain; mcgutierrez@icmm.csic.es (M.C.G.); delmonte@icmm.csic.es (F.d.M.)

**Keywords:** deep eutectic solvents, synthesis, ultrasound method, heating–stirring method

## Abstract

Deep eutectic solvents (DESs) have emerged as a greener alternative to other more polluting traditional solvents and have attracted a lot of interest in the last two decades. The DESs are less toxic dissolvents and have a lower environmental footprint. This paper presents an alternative synthesis method to the classical heating–stirring method. The ultrasound method is one of the most promising synthesis methods for DESs in terms of yield and energy efficiency. Therefore, the ultrasound synthesis method was studied to obtain hydrophobic (Aliquat 336:L-Menthol (3:7); Lidocaine:Decanoic acid (1:2)) and hydrophilic DESs based on choline chloride, urea, ethylene glycol and oxalic acid. The physical characterization of DESs via comparison of Fourier transform infrared (FTIR) spectra showed no difference between the DESs obtained by heating–stirring and ultrasound synthesis methods. The study and comparison of all the prepared DESs were carried out via nuclear magnetic resonance spectroscopy (NMR). The density and viscosity properties of DESs were evaluated. The density values were similar for both synthesis methods. However, differences in viscosity values were detected due to the presence of some water in hygroscopic DESs.

## 1. Introduction

In recent years, scientific research and industry have made significant efforts to find new, less polluting solvents to minimize the environmental impact, aiming to adhere to the principles of green chemistry.

Ionic liquids are considered an environmentally friendly alternative to traditional organic solvents; however, their poor biodegradability, biocompatibility and bio-sustainability, as well as the high use of organic solvents for their synthesis, minimize their large-scale use and increase their cost. Deep eutectic solvents (DESs) have emerged as a greener alternative to ionic liquids since 2003 [1]. DESs share general characteristics with ionic liquids, such as thermal stability, low volatility, low vapor pressure and tunable polarity; however, they offer advantages over ionic liquids, such as lower cost, biodegradability, lower toxicity and simpler production processes [2]. This is the reason why DESs have attracted so much interest in the last two decades, expanding their application in fields such as nanotechnology [3], food industry [4], medicine [5] and extraction [6,7]. 

DESs are obtained from the mixture of a hydrogen-bond donor (HBD) and a hydrogen-bond acceptor (HBA) [7]. The hydrogen bonding interactions allow for a very marked decrease in the freezing point. Martins et al. [8] defined the DES as a mixture of pure compounds for which the eutectic point temperature is below that of an ideal liquid mixture. The temperature depression should be defined as the difference between the ideal and the real eutectic point and should ensure that the mixture remains liquid for a certain composition range at the operating temperature. Their work is a study to establish the foundations to define, interpret and understand the DES denomination from a thermodynamic point of view. Otherwise, it suggests the fundamental properties needed and the approaches needed to guarantee that a certain mixture can really be considered a DES. As demonstrated, a DES is not a new pure compound but a mixture, which can be modified—for instance, by adding water—to improve its physicochemical properties and its liquid state temperature range. The synthesis methods of DESs are simple. They do not require multiple steps or separation methods or a purification step, since no solvents are needed [9]. The most commonly used methods are high-temperature (>50 °C) heating or grinding of the reagents.

The high-temperature heating method consists of mixing the reagents and heating at temperatures between 50 and 100 °C under stirring until a homogeneous liquid is formed [10,11]. It is necessary to identify the appropriate synthesis temperature, which is selected according to the boiling point and stability of the reagents. One of the most commonly used families of DESs is derived from the combination of carboxylic acids and choline chloride (ChCl). However, conventional heating, in the above range of temperature, can lead to degradation of DESs due to esterification reactions, in general, between the alcohol moiety of ChCl and the carboxylic acids [12,13]. 

The grinding method was carried out by Florindo et al. [12] for the preparation of DESs based on ChCl and carboxylic acids. The grinding method involves mixing the components and then grinding them in a mortar with a pestle at room temperature under nitrogen atmosphere or in a glovebox until a homogeneous liquid is obtained. This method is applicable to all solid components unsuitable for direct heating to prevent the formation of by-products. However, the quality of the generated DESs could be compromised due to the difficulty in achieving complete mixing.

Ultrasound or microwaves can be used to accelerate the synthesis process [14,15]. Stoichiometric amounts of HBD and HBA are mixed in a glass vial, sealed and placed in an ultrasound bath. The time and temperature required for DESs formation are based on the properties of pure constituents. 

The twin-screw extrusion technique serves as an alternative approach for DES preparation, addressing the constraints posed by conventional heating–stirring methods [11,16]. This method employs a setup comprising two screws rotating in opposing directions within a stainless-steel barrel, featuring multiple transport and kneading segments. Within the transport segment, materials are propelled forward, while in the kneading segment, high shear and compression forces act upon the material as it progresses. Following the preheating of the dual screw sections, HBA and HBD are integrated in suitable proportions.

Other synthesis methods are based on freeze-drying, which involves the dissolution of DES components separately in bi-distilled water to be subsequently mixed, frozen and freeze-dried until a clear viscous liquid is formed [17,18]. This is particularly suitable for the preparation of DESs with unstable thermal compounds that are water-soluble [15]. 

The vacuum evaporation method uses relatively lower temperatures compared to the traditional method of synthesizing DESs by heating–stirring. This synthesis method, first used for the preparation of natural DESs (NADES) by Dai et al. [19,20], consists of dissolving the DES components in water and then subjecting them to evaporation at 323 K, after which the DES is stored in a silica gel desiccator.

The microwave irradiation method often emerges as the more cost-effective and swifter option for synthesizing DESs [15]. In a groundbreaking study, Gomez et al. [21] unveiled a novel microwave-assisted technique, dramatically decreasing the synthesis time from 60 min to a mere 20 s while also achieving a remarkable 650-fold reduction in energy consumption. This method’s efficacy relies heavily on the careful calibration of heating duration, power levels and component selection. By prioritizing both time efficiency and energy conservation, Gomez et al. [21] pioneered a more environmentally friendly approach to preparing NADESs through microwave assistance.

Pelosi et al. [22] prepared DESs made from ChCl MgCl_2_·6H_2_O or CaCl_2_·6H_2_O, using a microwave coaxial antenna as a rapid, efficient and eco-friendly heating probe. This method can reduce the preparation time by up to 40 times compared to the conventional heating–stirring method without altering the final properties of the product. Rivera et al. [23] stated that the coaxial microwave technology is a mature, eco-friendly, cheap and easily implementable alternative to the classical microwave-oven-type reactors for preparing DESs.

Santana et al. [24] compared three different methods of synthesis of NADESs based on malic acid, citric acid and water, as well as xylitol, malic acid or citric acid and water, in each case using a molar ratio of 1:1:10. The synthesis methods included heating–stirring, ultrasonic-assisted and microwave-assisted methods. The resulting NADESs exhibited similar physicochemical properties. The synthesis conditions for traditional synthesis methods were 2 h at 323 K under magnetic stirring at 220 rpm. The ultrasound-assisted synthesis was carried out in an ultrasonic bath for 45 min, while the microwave synthesis was operated for 45 min at 353 K under a pressure of 10 bar. Time reduction is possible using ultrasound and microwave-assisted synthesis. Considering the synthesis time, the volume of solvent synthesized and the equipment power, the electric energy consumption for the DES synthesis was 0.014 kWh/mL for heating–stirring, 0.106 kWh/mL for microwave-assisted and 0.006 kWh/mL for ultrasound-assisted methods. Therefore, ultrasound and heating–stirring methods were faster and more efficient in terms of energy, especially the ultrasound method. Dlugosz et al. [25] compared microwave and ultrasound as alternative synthesis methods to obtain ChCl:urea (ChCl:U), ChCl:ethylene glycol (ChCl:EG), ammonium formate:EG and ammonium formate:glucose DESs with a conventional heating method. An increase in the yield is observed from 25 g/h for systems using conventional heating to as much as 800 g/h for systems obtained using non-conventional methods. The heating efficiencies of the microwave and ultrasound methods are 74.5% and 53.3%, respectively, compared to conventional heating, which is 5.0%.

Among the methods for obtaining DESs described in the literature, the ultrasound method is one of the most promising in terms of yield and energy. Therefore, this research work aims to obtain hydrophobic and hydrophilic DESs using the ultrasound method. The DESs obtained are compared with those obtained by a conventional heating–stirring method. The obtained DESs are characterized via Fourier transform infrared (FTIR) spectra. The density and viscosity of the mixtures obtained by both methods are compared. The DESs synthesized by the two methods are characterized via nuclear magnetic resonance (NMR) spectroscopy to confirm the stoichiometry of the mixtures.

## 2. Results and Discussion

The most common method of synthesis of DESs is heating and stirring the combination of the components until a clear liquid mixture is obtained. The purpose of this article is to analyze and discuss two methods of DESs preparation. Hydrophilic and hydrophobic DESs were synthesized using different methods according to the conditions given in Table 1 until homogeneous transparent liquids (without precipitation) were formed at room temperature. Some authors indicate that in the case of microwave or ultrasound energy, it is necessary to provide a semi-liquid or liquid reaction in order to conduct the energy of the ultrasonic waves [25,26,27]. However, in this work, DESs are synthesized using both methods, regardless of the physical state of their individual components. The synthesis times and temperatures for obtaining DESs are lower for the ultrasound method than for the traditional one. The decrease in the reaction time is really significant in the systems based on liquid–solid components (ChCl:EG) compared with systems where the raw materials are both solids (ChCl:U).

The formation of DESs is associated with the creation of hydrogen bonds between the precursors, a hydrogen-bond acceptor (HBA) and a hydrogen-bond donor (HBD), along with intermolecular interactions involving Van der Waals and electrostatic forces. FTIR is an important tool for analyzing the supramolecular structure of the obtained DESs and for elucidating the effects of modifications in the processing conditions in terms of the formation of functional groups and changes in composition [28]. The FTIR spectra obtained for the DESs from different synthesis methods are shown in Figure 1, Figure 2, Figure 3, Figure 4 and Figure 5. A comparison of the spectra showed no difference in the DES, regardless of the synthesis method. The FTIR spectra of DESs are a combination of the spectra of their starting compounds, except for a few frequency shifts associated with the formation of hydrogen bonds. No degradation of the samples was observed.

Figure 1 depicts the spectra of the ChCl:U (1:2) DESs synthesized by both methods, showing that both spectra are similar. The urea absorption bands at 3429 cm^−1^, 3336 cm^−1^ and 3256 cm^−1^ are assigned to νas NH_2_, νs NH_2_ and δs NH_2_. After the formation of the DESs, broader bands are observed at shorter wavelengths, between 3319 cm^−1^ and 3192 cm^−1^. This may be due to the formation of H- bonds between urea and ChCl. The shifting of the signals at 1674 cm^−1^ and 1620 cm^−1^ assigned to δs NH_2_ and δas NH_2_ at the values of 1662 cm^−1^ and 1608 cm^−1^ is also associated with the formation of strong H-bonds [29]. Also, a shift to lower wavenumbers of the 1460 cm^−1^ signal assigned to ρs NH_2_ to values around 1435 cm^−1^ is observed. The band at 3220 cm^−1^ of ChCl assigned to the ν_sm_ of OH disappears in the DESs [28]. The signals at 1474 cm^−1^ and 958 cm^−1^ are assigned to the N–C bond and the νCCO of ChCl, respectively, indicating that the ChCl structure has not been destroyed in the formation of the DES [29,30].

Figure 2 shows the FTIR spectra obtained for ChCl:EG DESs synthesized from different ratios: (1:2) (a), (1:3) (b) and (1:4) (c). Again, the FTIRs obtained for these DESs synthesized via the heating–stirring or ultrasound methods are similar. In all of them, the formation of the DESs is associated with the observed shift in the νas OH signal from 3304 cm^−1^ in EG to the range of 3316–3309 cm^−1^ in the synthesized DESs. This shifting is due to a loss of the crystalline structure associated with the formation of H-bonds [31,32,33,34]. The signal assigned to the νsm of the OH of ChCl (3220 cm^−1^) disappears in the DESs. In all the ChCl:EG DESs, similar signals to those of the starting compounds are observed, regardless of the ChCl:EG ratio used. Therefore, the bands at 2938 cm^−1^ and 2875 cm^−1^ correspond to the νas and νs of the C–H, and the signals at 1084 and 1037 cm^−1^ correspond to the νas and νs of the C–O, as well as those at 883 and 863 cm^−1^ assigned to *ρ* CH_2_ and ν C–C [35,36]. Again, the signals around 1482 cm^−1^ and 958 cm^−1^, assigned above to the C–N bond and the νCCO of ChCl, respectively, indicate that the ChCl structure has not changed in the formation of the DESs. No changes are observed in the spectra with an increasing EG ratio.

Figure 3 shows the FTIR spectrum obtained for ChCl:oxalic acid (ChCl:Ox 1:1). The spectrum of oxalic acid shows broad overlapping bands centered at 3450 cm^−1^ in the –OH stretching region typical of carboxylic acid forming dimer rings strongly linked through intermolecular H-bonds and H-bonds between the C=O and O–H groups. It presents a band between 1668 and 1614 cm^−1^ assigned to the C=O stretching frequency; its splitting is interpreted by the coupling of slightly different C=O groups [37]. Around 1440 cm^−1^, a signal assigned to C–O stretching appears. These last two signals can be used to recognize the presence of free acid in the synthesized DESs [28]. The DESs obtained using heating–stirring as well as ultrasound methods show similar FTIR spectra. These correspond to a combination of the spectra of the starting compounds with a shift in certain signals. A broadening and shifting to lower wavenumbers of the signal assigned to the OH stretching are observed around 3325 cm^−1^; the band assigned to the OH group of ChCl at 3220 cm^−1^ disappears. Signals at 1720 cm^−1^ indicate the presence of free carbonyl groups (COOH), and the signal at 1633 cm^−1^ indicates the presence of dissolved Ox [28]. As in the previous DESs, the bands at 1482 cm^−1^ and 958 cm^−1^ are assigned to the C–N bond and the νCCO of ChCl, respectively, and the bands at 1184 and 1084 cm^−1^ correspond to the νas and νs of the C–O [28,30,38]. 

The FTIR spectra for the hydrophobic compounds are presented in Figure 4 and Figure 5. The FTIR spectra of the DESs Aliquat 336: L-menthol (Aliq:L Met 3:7) obtained using both methods are similar and present the signals of the starting compounds (see Figure 4). At 3362 cm^−1^, bands assigned to the hydroxyl group are observed; between 2855 cm^−1^ and 2924 cm^−1^, signals assigned to the νas and νs of the C–H appear; and at 1025 cm^−1^ and 1045 cm^−1^, signals attributed to the stretching vibration of the C–O group appear, with the band at 1368 cm^−1^ corresponding to the isopropyl group of L Met. [39]. In the region 1460–1463 cm^−1^, signals assigned to the quaternary amine (CH_3_)_3_N^+^ appear, and the region between 2922 cm^−1^ and 3028 cm^−1^ corresponds to the asymmetric bands of the C–H [40,41]. The formation of DESs is confirmed by the broadening of the signal assigned to the OH group (3326 cm^−1^) and its displacement to higher wavenumbers (3326 cm^−1^), which indicates that the formation mechanism corresponds to the intermolecular interaction of the hydroxyl groups of L Met with the Cl^−^ of Aliq. A displacement of the signals appearing at 1047 cm^−1^ and 1028 cm^−1^ is also observed, which is assigned to the stretching vibration of the C–O group due to a change in the state of aggregation associated with the formation of H-bonds [34]. 

The FTIR spectra obtained for the DESs Decanoic Ac:Lidocaine (Lid:Ac. Dec (1:2)) using the heating–stirring or ultrasound synthesis methods are similar (see Figure 5). The DES signals assigned to the starting compounds are observed. The formation of DESs is associated with the presence of a signal centered at 1550 cm^−1^, which is not present in the starting compounds. This vibrational mode corresponds to the asymmetric stretching vibration of the carboxylate ion, i.e., the deprotonated Ac. Dec. In addition, a signal is observed at 1690 cm^−1^, which is related to the strong hydrogen bonding that occurs between the carboxyl group of Decanoic acid and the tertiary amino group of Lidocaine [42]. 

The following figures (Figure 6, Figure 7, Figure 8 and Figure 9) and tables (Table 2, Table 3, Table 4, Table 5 and Table 6) show the NMR spectra and proton assignments, respectively, of all DESs prepared via the heating–stirring and ultrasound methods. The acquired spectra revealed the presence of DES components in the respective molar ratios they were combined in for DES formation. Moreover, DESs composed of mixtures of carboxylic acids and ChCl were found to evolve into the respective ester derivatives via esterification reaction between the alcohol moiety of ChCl and carboxylic acid. Rodriguez Rodriguez et al. [13] described significant esterification of Ox upon initial DES preparation at 60 °C, around 11 mol% of ChCl, as we also observed in this work for this acidic DES, regardless of the methodology used (i.e., heating–stirring or ultrasound). The final temperature attained in the ultrasound bath during the synthesis (ca. 58 °C; see Table 1) is close to the temperature used in the heating–stirring method (i.e., 60 °C). This similarity in temperatures could explain the insignificant differences found in the esterification percentages of Ox detected in the DESs prepared using both methods (Figure 6 and Table 4).

Table 2 shows the assignment of the peaks corresponding to each NMR spectrum. A signal for water protons is observed in the spectra of hydrophilic DESs (Table 2, Table 3 and Table 4).

Table 6 includes the chemical shifts in the hydrophobic DES Aliq:L Met (3:7) and its individual components. The formation of H-bond complexes in this DES was confirmed by the upfield chemical shift in the signals ascribed to Aliq and L Met in the ^1^H NMR spectra of the DESs (both synthesized using heating–stirring or ultrasound methods) as compared to that of the individual components (see, for example, ^+^N–CH_3_ shifts from 2.94 ppm in Aliq spectrum to 2.88 ppm after DES formation or H1 shifts from 2.87 ppm in L Met spectrum to 2.74 ppm in the spectra of DES). The characteristic peak of the hydroxyl group in the L Met spectrum shifts from 3.52 to 4.05 ppm after DES formation in agreement with the presence of H-bonds in Aliq:L Met (3:7) DESs [34].

To study the effect of the synthesis method on the properties of the DESs, the density and viscosities were evaluated. Both properties are determinants in the application of DESs. The density values of DESs oscillate between 1.0 and 1.3 g cm^−3^ at 25 °C [7]. Table 7 shows the densities for the DESs synthesized using the different methods. The results obtained show that the densities are similar, indicating that the synthesis method does not influence the density of the compounds obtained.

Appendix A shows bibliographic results of densities for the different DESs. The densities obtained are similar to the bibliographic ones; the slight differences observed may be due to the methods used to measure the densities or the presence of impurities in the compounds. 

DES viscosity is a fundamental parameter that influences the transport properties and can be used to define their potential applications [7]. DESs are usually quite viscous fluids compared with organic solvents. Viscosity changes significantly as a function of temperature, the H-bond donor type and the mixture composition. Table 8 shows the viscosity values of the DESs prepared using the two methods at different temperatures. 

It is observed that the viscosity decreases with increasing temperature in all the DESs obtained, regardless of the synthesis method. This behavior is explained using the hole theory. Abbott and co-workers first applied this theory to DESs [43,44]. Viscosity and electrical conductivity depend on the presence of holes in the liquids that facilitate the mobility of the compounds in the final network. 

These holes, of different sizes and locations, are in continuous movement. At lower temperatures, the size of the holes is small compared to the size of the DES components, which therefore fit with difficulty into the holes, reducing the free mobility of the components and increasing the viscosity of the system. This behavior is reversed at high temperatures; the average hole size increases with temperature and is comparable to the size of the components, thus increasing their mobility [45,46]. According to this theory, viscosity is controlled by volumetric factors or steric effects that influence the intermolecular interactions between HBA and HBD rather than by the stronger interaction between the two components. 

In any case, it is important to keep in mind that the hole theory is just a theory that may apply to some systems but not to others.

The results obtained show differences in the viscosity values of hydrophilic DESs depending on their synthesis method (Table 8). In general, they are higher for the DES prepared using the heating–stirring method than for the corresponding DES prepared using the ultrasound method. The main differences are obtained for ChCl:U (1:2) and ChCl:Ox (1:1). 

The viscosity differences could be ascribed to a higher water content in the samples obtained using the ultrasonic-assisted method [12,47], according to analysis of the water present in each DES (see Table 9). Higher water contents imply a decrease in the H-bonds between the HBA and HBD of the DES [48].

Table 9 shows the water content in the DESs prepared using the two methods studied.

It is observed that the water content in hydrophilic DESs is higher than in hydrophobic DESs. In the case of hydrophilic DESs, the water content is higher in those prepared using the ultrasound method. Based on the results, it can be inferred that water contents are associated with the viscosity differences observed in the DESs prepared using the two methods studied (see Table 8).

Based on the data of viscosity, it can be seen that the behavior is different for hydrophobic DES. The viscosity values are slightly higher for DESs prepared using the ultrasound method than those DESs prepared using the heating–stirring method.

It is observed that in the case of DESs ChCl:EG prepared with different molar ratios (1:2, 1:3 and 1:4), the viscosity generally decreases with an increasing concentration of the donor compound (EG) [46]. Low viscosity is beneficial for mass transfer between the solute and the solvent [49]. This decrease in viscosity is an interesting property for achieving low-viscosity metal leachates, allowing for problem-free handling.

The model commonly used to study the effect of temperature on the viscosity of DESs is the Arrhenius model. This equation gives a correlation between viscosity and temperature:ln η = ln η_0_ + E_η_/RT(1)
where ln η is the logarithm of viscosity; η_0_ is a pre-exponential constant; E_η_ is the energy for the activation of the viscous flow; R is the universal gas constant; and T is the temperature. 

Figure 10 shows that all of the data obey Equation (1) well (R^2^ ≥ 0.99) (Figure 10a: heating–stirring method) (R^2^ ≥ 0.99) (Figure 10b: ultrasonic-assisted synthesis). 

Table 10 shows the activation energy values obtained for each DES and the corresponding values of ⴄ_0_. Less viscous DESs show low Ea values, such as ChCl:EG (1:3), whereas highly viscous DESs have larger Ea, such as ChCl:U (1:2). It is observed that the DESs that present the highest activation energy are those obtained using the heating–stirring method. Hence, the calculated data of Ea perfectly correlate with the measured values of viscosity. 

The activation energy of DESs formed by ChCl and EG in the three molar ratios studied (1:2, 1:3 and 1:4) presents lower energy, independently of the synthesis method used. The DESs that show the greatest difference in the activation energy according to the synthesis method used are ChCl:U 1:2. E_η_ = 77.90 KJ/mol using the heating–stirring method and E_η_ = 50.75 KJ/mol using the ultrasonic-assisted synthesis.

Figure 11 shows the DSC plots of DESs synthesized using both heating–stirring and ultrasonic-assisted methods. No melting points were observed in any of the DSC curves. The presence of water, detected using Karl Fisher measurements, could explain this result, as previously shown by Jany et al. for DESs based on ChCl:EG [50]. A glass transition temperature (Tg) was observed only for ChCl:U (1:2) DESs with Tg values of −68.10 °C and for ChCl:U (1:2) DESs with Tg values of −66.13 °C synthesized using the heating–stirring and ultrasound methods, respectively. The different water contents of these U-based DESs could also explain the variations in Tg values.

## 3. Materials and Methods

### 3.1. Materials and Reagents

Hydrophilic and hydrophobic DESs were synthesized using choline chloride (ChCl, high purity grade), urea > 99 (N), pure ethylene glycol (EG), oxalic acid 2-hydrate (Ox) 99%, L-menthol (L Met) 99%, Aliquat 336 TG (Aliq), lidocaine (Lid) 97.5% and decanoic acid (Ac. Dec) 99%. The description of the chemical reagents used for synthesis of the DESs is shown in Table 11. The proportions for the synthesis of each of the DESs are shown in Table 12.

### 3.2. Synthesis Method

The synthesis of DESs using the heating–stirring method was carried out by mixing the two respective components under continuous stirring (at 600 rpm) at a fixed temperature. The components of each DES were placed in an Erlenmeyer flask and immersed in a silicone oil bath with immersion thermostat (model DIGITERM TFT-200, J.P. SELECTA, Scharlab S.L., Barcelona, Spain). 

The ultrasound synthesis of DESs was carried out in an ultrasound bath (A&J Tecno Ultrasound, Paterna, Spain). The respective amount of components was mixed in an Erlenmeyer flask until a homogeneous liquid was obtained. The prepared DESs were transferred to sealed flasks.

### 3.3. Physical–Chemical Characterization of the DESs

The infrared spectra of the DESs synthesized using the two methods studied, as well as the initial reagents, were obtained using a Nicolet iS50 FT-IR Spectrometer from Thermo Scientific™ (Waltham, MA, USA) operated in the attenuated total reflectance (ATR) mode, in the range of 4000–400 cm^−1^, with a spectral resolution of 4 cm^−1^ and accumulation of 64 scans.

^1^H NMR spectra of the samples obtained using the heating–stirring and ultrasonic methods were conducted using a Bruker Avance DRX500 spectrometer (Billerica, MA, USA) operating at 500 MHz. The NMR parameters employed included a 30° pulse, an acquisition time of 3.1719 s, a relaxation delay of 1 s and a total of 16–32 scans. All samples were placed in capillary tubes inside 5 mm NMR glass tubes and analyzed using deuterated chloroform (CDCl_3_) or dimethyl sulfoxide (DMSO-*d*_6_) as an external reference. The spectra were acquired after setting the temperature to 298 or 333 K using a Bruker Variable Temperature BVT 3000.

Measurements of the densities of DESs were acquired based on the difference in weight using an Ohaus Explorer Analytical 110G Balance from OHAUS Europe GmbH (Nänikon, Switzerland), for which 6 measurements were carried out in which the standard deviations of the samples were calculated. 

The viscosities of DESs were determined using a viscometer PCE-RVI2 (Meschede, Germany). We admitted that the studied DESs exhibit a Newtonian behavior. Elhamarnah et al. [51] state that the research conducted on NADESs has shown that they behave primarily like Newtonian fluids. The analyses were performed at different temperatures (20–70 °C). Temperature control was carried out in a water bath. Four measurements were carried out in which the standard deviation and error of the samples were calculated.

The water content in the prepared DESs was determined using a Karl Fischer C20/C30 titrator from Mettler Toledo (Singapore). Five measurements were taken for each DES, and the mean and standard deviation of these measurements were calculated.

DSC scans were performed in a TA Instruments Model Q2000 (New Castle, DE, USA). Samples were placed into an aluminum pan in a sealed furnace and cooled to −90 °C at a scan rate of 10 °C min^−1^, performing two heating-and-cooling cycles at the same scan rate.

## 4. Conclusions

The ultrasound method is one of the most promising methods for synthesizing DESs in terms of yield and energy efficiency. According to the results, it is possible to obtain both hydrophilic (e.g., ChCl:U (1:2), ChCl:Ox (1:1), ChCl:EG (1:2), (1:3) and (1:4)) and hydrophobic (e.g., Lid:Ac. Dec (1:2), Aliq:L Met (3:7)) DESs using the ultrasonic-assisted method, which results in lower reaction times and temperatures. The formation of DESs using both the heating–stirring and ultrasound methods was confirmed by the frequency shift in certain bands associated with the formation of H-bonds. The DESs composed of mixtures of carboxylic acids and ChCl were found to evolve into the respective ester derivatives. The synthesis method used did not affect the density values of DESs. However, a decrease in the viscosity of hydrophilic DESs obtained using the ultrasound method was observed, likely due to their higher water content compared to the same DESs prepared using the heating–stirring method. The Arrhenius model fits the correlation between viscosity and temperature.

## Figures and Tables

**Figure 1 molecules-29-03089-f001:**
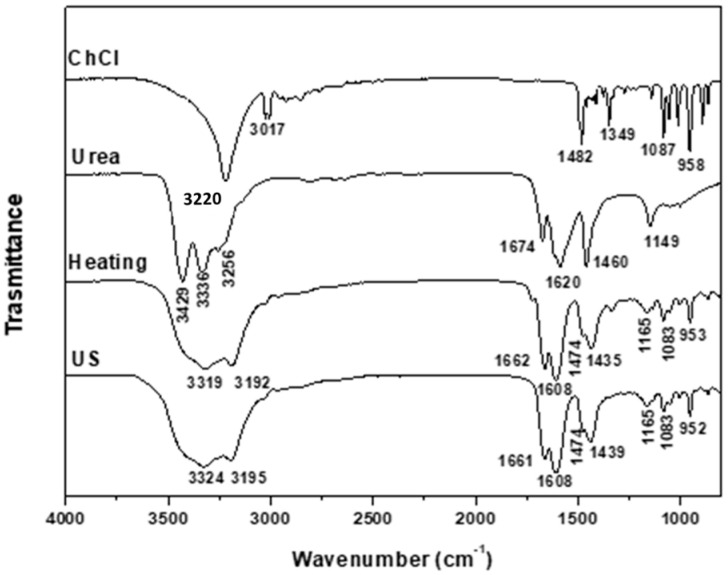
FTIR spectra of ChCl, urea and ChCl-urea (1:2) DESs obtained using the heating–stirring and ultrasound (US) methods.

**Figure 2 molecules-29-03089-f002:**
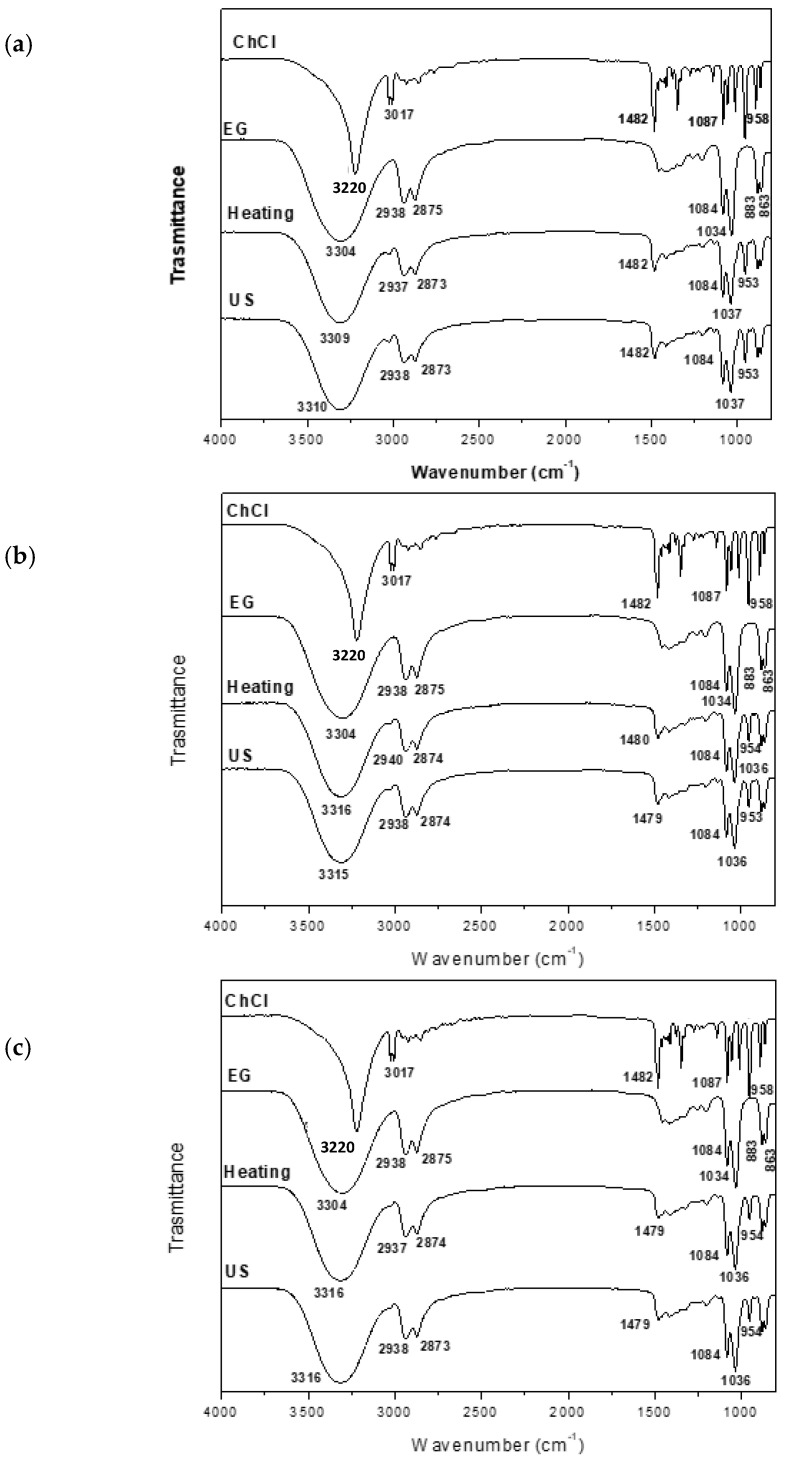
FTIR spectra of DESs (**a**) ChCl:EG (1:2), (**b**) ChCl:EG (1:3) and (**c**) ChCl:EG (1:4).

**Figure 3 molecules-29-03089-f003:**
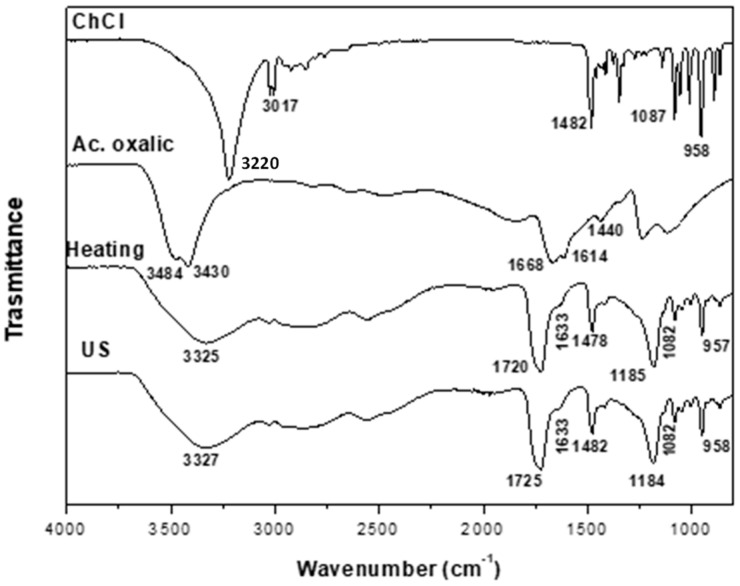
FTIR spectra of ChCl, oxalic acid and ChCl:Ox DESs (1:1) obtained using the heating–stirring and ultrasound methods.

**Figure 4 molecules-29-03089-f004:**
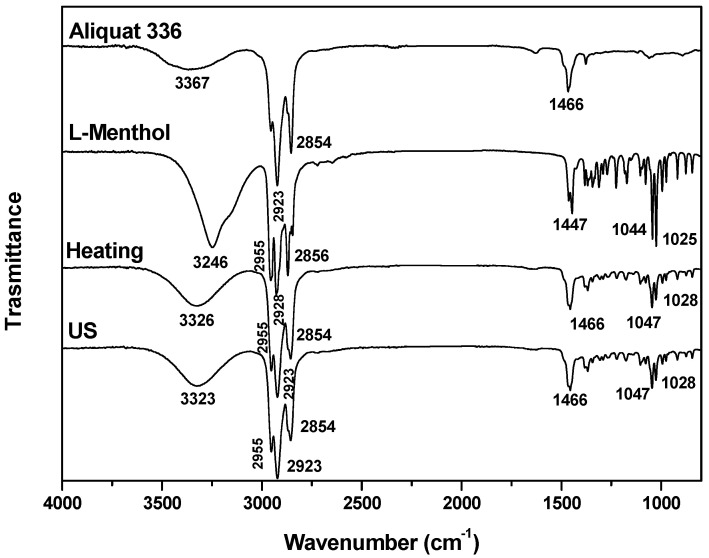
FTIR spectra of L-menthol, Aliquat 336 and Aliq:L Met (3:7) DESs obtained using the heating–stirring and ultrasound methods.

**Figure 5 molecules-29-03089-f005:**
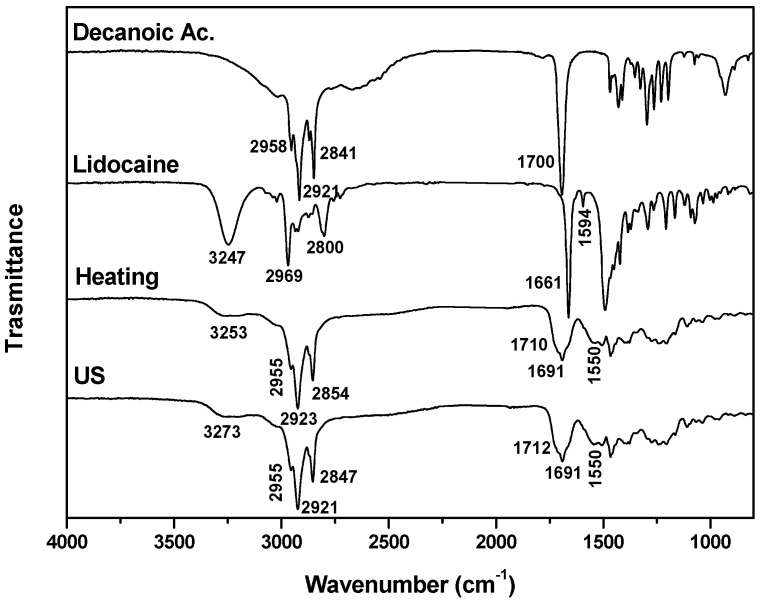
FTIR spectra of Decanoic ac, lidocaine and Lid:Ac. Dec (1:2) DESs obtained using the heating–stirring and ultrasound methods.

**Figure 6 molecules-29-03089-f006:**
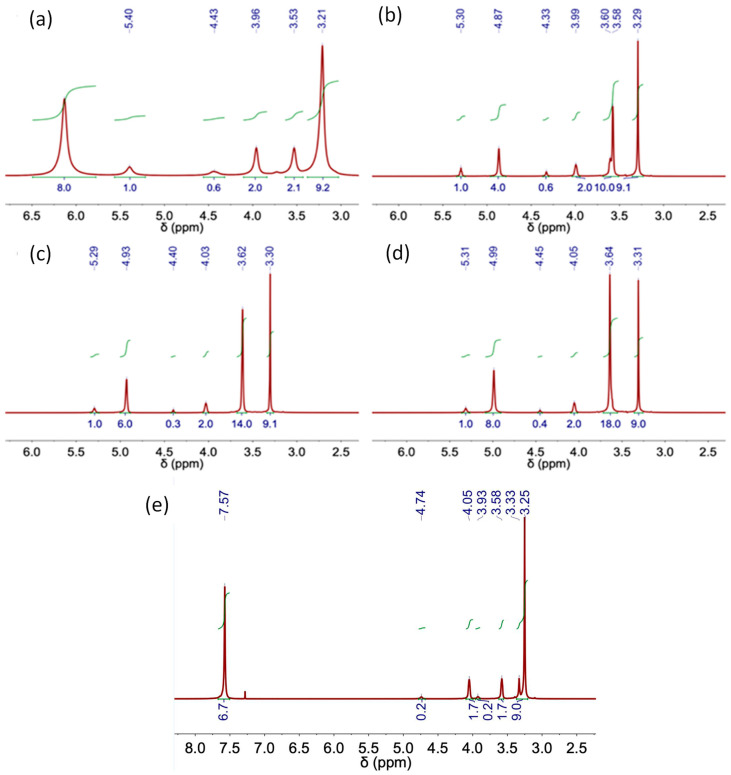
NMR spectrum of DES obtained using the heating–stirring method. (**a**) ChCl-Urea (1:2), (**b**) ChCl:EG (1:2), (**c**) ChCl:EG (1:3), (**d**) ChCl:EG (1:4) and (**e**) ChCl:Ox (1:1).

**Figure 7 molecules-29-03089-f007:**
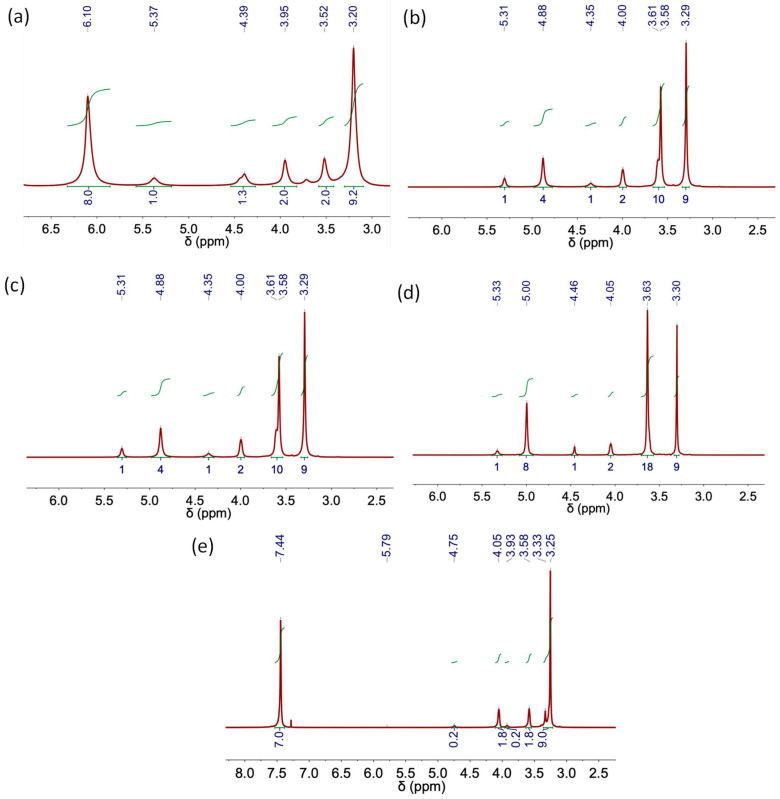
NMR spectrum of DES obtained using the ultrasound method. (**a**) ChCl-Urea (1:2), (**b**) ChCl:EG (1:2), (**c**) ChCl:EG (1:3), (**d**) ChCl:EG (1:4) and (**e**) ChCl:Ox (1:1).

**Figure 8 molecules-29-03089-f008:**
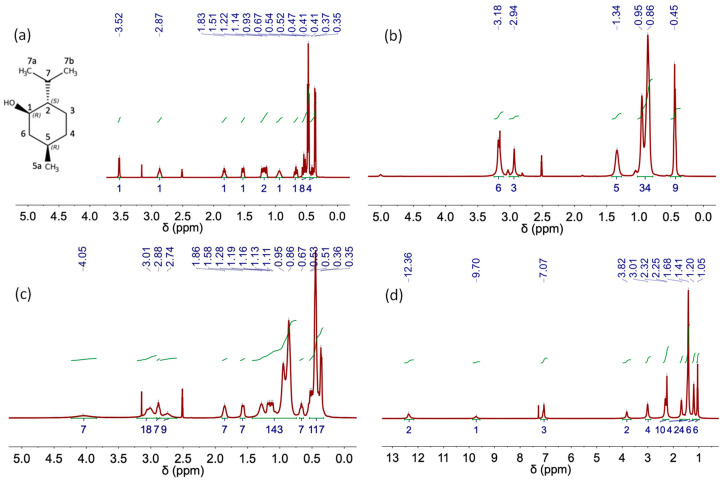
NMR spectra of (**a**) L-Menthol, (**b**) Aliquat 336 TG, (**c**) Aliq:L Met (3:7) and (**d**) Lid:Ac. Dec (1:2) DES obtained using the heating–stirring method.

**Figure 9 molecules-29-03089-f009:**
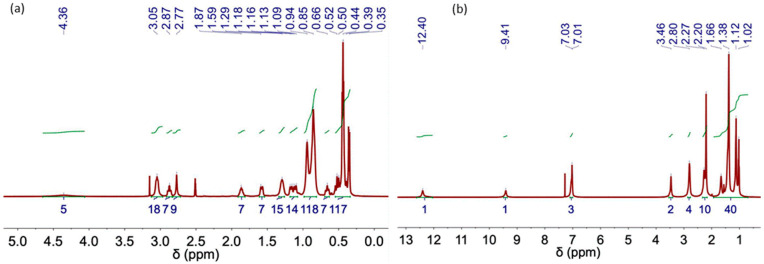
NMR spectra of (**a**) Aliq:L Met (3:7) and (**b**) Lid:Ac. Dec (1:2) DES obtained using the ultrasound method.

**Figure 10 molecules-29-03089-f010:**
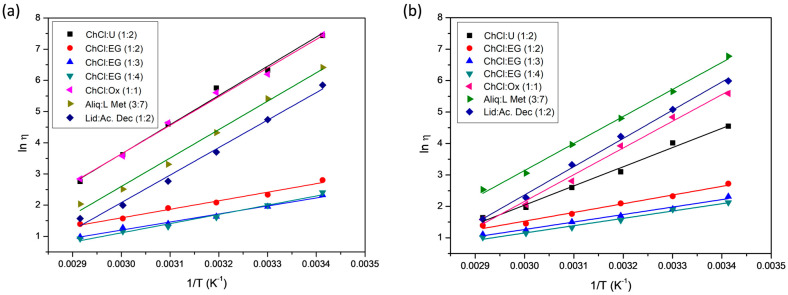
Plots of ln viscosity vs. inverse of temperature for different DESs synthesized using the two methods studied: (**a**) heating–stirring method, (**b**) ultrasonic-assisted synthesis.

**Figure 11 molecules-29-03089-f011:**
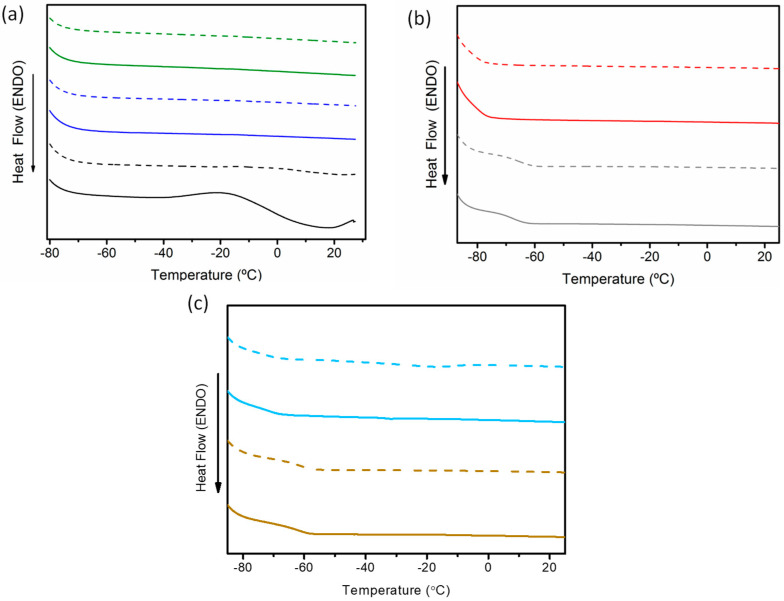
DSC curves obtained from the studied DESs synthesized via heating–stirring (solid lines) and ultrasound (dashed lines) methods. (**a**) ChCl:EG (1:2) (black), ChCl:EG (1:3) (purple), ChCl:EG (1:4) (green); (**b**) ChCl:U (1:2), (gray), ChCl:Ox (red); and (**c**) Aliq:L Met (3:7) (light blue), Lid:Ac. Dec (1:2) (brown).

**Table 1 molecules-29-03089-t001:** Synthesis conditions for each DES prepared by heating–stirring and ultrasound methods.

DES	Heating–Stirring		Ultrasound	
	Time (h)	Temperature (°C)	Time (h)	Final Temperature * (°C)
ChCl:U (1:2)	5	80	4	50
ChCl:EG (1:2)	24	80	1	50
ChCl:EG (1:3)	24	80	1	45
ChCl:EG (1:4)	24	80	1	45
ChCl:Ox (1:1)	4.5	60	3	58
Aliq:L Met (3:7)	0.5	60	0.5	46
Lid:Ac. Dec (1:2)	1	60	1	44

* Final temperature attained spontaneously.

**Table 2 molecules-29-03089-t002:** ^1^H NMR chemical shifts in hydrophilic DESs of (ChCl) and (EG) obtained using the heating-stirring and ultrasound (US) methods. In this case, the spectra were recorded at 298 K using CDCl_3_ as the external reference.

δ (ppm)							
DES	**EG**		**ChCl ***				**H**_2_O **^a^**
	O**H ^a^**	O–C**H**_2_ **^a^**	O**H ^a^**	O–C**H**_2_ **^a^**	^+^N–C**H**_2_ **^a^**	^+^N–(C**H**_3_)_3_ **^a^**	
ChCl:EG (1:2)	4.87 (4H)	3.60–3.58 (**8H** + 2H) **^b^**	5.30 (1H)	3.99 (2H)	3.60–3.58 (**2H** + 8H) **^c^**	3.29 (9H)	4.33 (0.6H)
ChCl:EG (1:2) US	4.88 (4H)	3.61–3.58 (**8H** + 2H) **^b^**	5.31 (1H)	4.00 (2H)	3.60–3.58 (**2H** + 8H) **^c^**	3.29 (9H)	4.35 (0.7H)
ChCl:EG (1:3)	4.93 (6H)	3.63 (**12H** + 2H) **^b^**	5.29 (1H)	4.03 (2H)	3.62 (**2H** + 12H) **^c^**	3.30 (9H)	4.40 (0.3H)
ChCl:EG (1:3) US	4.93 (6H)	3.61 (**12H** + 2H) **^b^**	5.29 (1H)	4.03 (2H)	3.61 (**2H** + 12H) **^c^**	3.30 (9H)	4.40 (0.7H)
ChCl:EG (1:4)	4.99 (8H)	3.64 (**16H** + 2H)** ^b^**	5.31 (1H)	4.05 (2H)	3.64 (**2H** + 16H) **^c^**	3.31 (9H)	4.45 (0.4H)
ChCl:EG (1:4) US	5.00 (8H)	3.63 (**16H** + 2H) **^b^**	5.33 (1H)	4.05 (2H)	3.63 (**2H** + 16H) **^c^**	3.30 (9H)	4.46 (1H)

**^a^** Bold H indicates the protons in the compound to which the chemical shifts refer. **^b^** Bold numbers refer to the number of protons in the integral assigned to CH_2_ groups of EG and non-bold numbers refer to the number of protons in the integral assigned to the CH_2_ group bonded to the N of ChCl. **^c^** Bold numbers refer to the number of protons in the integral assigned to CH_2_ group bonded to the N of ChCl and non-bold numbers refer to the number of protons in the integral assigned to CH_2_ groups of EG. * The asterisk indicates the protons in the choline oxalate.

**Table 3 molecules-29-03089-t003:** ^1^H NMR chemical shifts in hydrophilic DES of (ChCl) and (U) obtained using the heating-stirring (upper row) and ultrasound (US) (lower row) methods. The spectra were recorded at 298 K using CDCl_3_ as the external reference.

δ (ppm)						
**DES**	**U**	**ChCl**				**H**_2_O **^a^**
	C–N**H**_2_ **^a^**	O**H ^a^**	O–C**H**_2_ **^a^**	^+^N–C**H**_2_ **^a^**	^+^N–(C**H**_3_)_3_ **^a^**	
ChCl:U (1:2)	6.13 (8H)	5.40 (1H)	3.96 (2H)	3.53 (2H)	3.21 (9H)	4.43 (0.6H)
ChCl:U (1:2) US	6.10 (8H)	5.37 (1H)	3.95 (2H)	3.52 (2H)	3.20 (9H)	4.39 (1.3H)

**^a^** Bold H indicates to the protons in the compound to which the chemical shifts refer.

**Table 4 molecules-29-03089-t004:** ^1^H NMR chemical shifts in hydrophilic DES of (ChCl) and (Oxalic acid) obtained from the heating-stirring (upper row) and ultrasound (US) (lower row) methods. The spectra were recorded at 298 K using CDCl_3_ as the external reference.

δ (ppm)									
**DES**	**Oxalic Acid**	**ChCl**							**H**_2_O **^a^**
	(COO**H**)_2_ **^a^**	O**H ^a^**	O–C**H**_2_ *^,^**^a^**	O–C**H**_2_ **^a^**	^+^N–C**H**_2_ *^,^**^a^**	^+^N–C**H**_2_ **^a^**	^+^N–(C**H**_3_)_3_ *^,^**^a^**	^+^N–(C**H**_3_)_3_ **^a^**	
ChCl:Ox (1:1)	7.57 (**2H** + 1H + 3.7H) **^b^**	7.57 (**1H** + 2H + 3.7H) **^c^**	4.74 (0.2H)	4.05 (1.7H)	3.93 (0.2H)	3.58 (1.7H)	3.33	3.25	7.57 (**3.7H** + 2H + 1H) **^d^**
							(9H)		
ChCl:Ox (1:1) US	7.44 (**2H** + 1H + 4H) **^b^**	7.44 (**1H** + 2H + 4H) **^c^**	4.75 (0.2H)	4.05 (1.8H)	3.93 (0.2H)	3.58 (1.8H)	3.33	3.25	7.44 (**4H** + 2H + 1H) **^d^**
							(9H)		

* The asterisk indicates the protons in the choline oxalate. **^a^** Bold H indicates the protons in the compound to which the chemical shifts refer. **^b^** Bold numbers refer to the number of protons in the integral assigned to acid groups of Ox and non-bold numbers refer to the number of protons in the integral assigned to the OH group of ChCl and to the H_2_O. **^c^** Bold numbers refer to the number of protons in the integral assigned to the OH group of ChCl and non-bold numbers refer to the number of protons in the integral assigned to acid groups of Ox and to the H_2_O. **^d^** Bold numbers refer to the number of protons in the integral assigned to the H_2_O and non-bold numbers refer to the number of protons in the integral assigned to acid groups of Ox and to the OH group of ChCl.

**Table 5 molecules-29-03089-t005:** ^1^H NMR chemical shifts in hydrophobic DES of Lid:Ac Dec (1:2) obtained from the heating-stirring (upper row) and ultrasound (US) (lower row) methods. The spectra were recorded at 298 K using CDCl_3_ as the external reference.

δ (ppm)										
Decanoic Acid					Lidocaine					
**H**OOC	HOOC–C**H**_2_ **^a^**	OC–CH_2_– C**H**_2_(CH_2_)_6_ **^a^**	(C**H**_2_)_6_ **^a^**	C**H**_3_ **^a^**	N**H ^a^**	Ar–**H ^a^**	Ar–C**H**_3_ **^a^**	OC–C**H**_2_–N **^a^**	C**H**_2_–CH_3_ **^a^**	CH_2_–C**H**_3_ **^a^**
12.36 (2H)	2.32 (**4H** + 6H) **^b^**	1.68 (4H)	1.41 (24H)	1.05 (6H)	9.70 (1H)	7.07 (3H)	2.25 (**6H** + 4H) **^c^**	3.82 (2H)	3.01 (4H)	1.20 (6H)
12.40 (2H)	2.27 (**4H** + 6H) **^b^**	1.66 (4H)	1.38 (24H)	1.02 (6H)	9.41 (1H)	7.03 (3H)	2.20 (**6H** + 4H) **^c^**	3.46 (2H)	2.80 (4H)	1.12 (6H)

**^a^** Bold H indicates the protons in the compound to which the chemical shifts refer. **^b^** Bold numbers refer to the number of protons in the integral assigned to the CH_2_ group bonded to the acid group of decanoic acid and non-bold numbers refer to the number of protons in the integral assigned to the CH_3_ groups bonded to the aromatic ring of lidocaine. **^c^** Bold numbers refer to the number of protons in the integral assigned to the CH_3_ groups bonded to the aromatic ring of lidocaine and non-bold numbers refer to the number of protons in the integral assigned to the CH_2_ group bonded to the acid group of decanoic acid.

**Table 6 molecules-29-03089-t006:** ^1^H NMR chemical shifts in L Met, Aliq and the hydrophobic DES Aliq:L Met (3:7) obtained from the heating-stirring and ultrasound (US) methods. The spectra were recorded at 333 K using deuterated DMSO as the external reference.

		δ (ppm)				
		L Met	Aliquat 336 TG	Aliq:L Met (3:7)	Aliq:L Met (3:7) US	
L-Menthol	H1	2.87 (1H)		2.88 (7H)	2.87 (7H)	Aliq-L Met (3:7)
	H2	0.67 (1H)		0.67 (7H)	0.66 (7H)	
	H3a	1.22 (1H)		1.17	1.17 (7H)	
	H3b	0.41 (1H)		0.53–0.35	0.52–0.35	
	H4a	1.14 (1H)		1.12	1.11 (7H)	
	H4b	0.54 (1H)		0.53–0.35	0.52–0.35	
	H5	0.93 (1H)		0.95–0.86	0.94–0.85	
	H6a	1.51 (1H)		1.58 (7H)	1.59 (7H)	
	H6b	0.52 (1H)		0.53–0.35	0.52–0.35	
	H7	1.83 (1H)		1.86 (7H)	1.87 (7H)	
	CH_3_ 5a	0.47 (3H)		0.45	0.44	
	CH_3_ 7a	0.36 (3H)		0.36	0.35	
	CH_3_ 7b	0.47 (3H)		0.45	0.44	
	OH	3.52 (1H)		4.05 (7H)	4.05 (5H)	
Aliquat-336 TG** ^a^**	^+^N–C**H**_3_		2.94 (3H)	2.74 (9H)	2.77 (9H)	
	^+^NC**H**_2_CH_2_(CH_2_)_n_CH_3_		3.18 (6H)	3.01 (18H)	3.05 (18H)	
	^+^NCH_2_C**H**_2_(CH_2_)_n_CH_3_		1.34 (5H)	1.28	1.29 (15H)	
	^+^NCH_2_CH_2_(C**H**_2_)_n_CH_3_		0.95–0.86 (34H)	0.95–0.86	0.94–0.85	
	^+^NCH_2_CH_2_(CH_2_)_n_C**H**_3_		0.45 (9H)	0.45	0.44	

**^a^** Bold H indicates to the protons of the CH_2_ groups in Aliquat-336 TG to which the chemical shifts refer.

**Table 7 molecules-29-03089-t007:** Densities of hydrophilic and hydrophobic DESs measured at 25 °C.

DES	Density (g/cm^3^) Heating–Stirring	Density (g/cm^3^) Ultrasound
Hydrophilic DES		
ChCl:U (1:2)	1.185 ± 0.006	1.165 ± 0.002
ChCl:EG (1:2)	1.093 ± 0.001	1.102 ± 0.005
ChCl:EG (1:3)	1.095 ± 0.002	1.105 ± 0.007
ChCl:EG (1:4)	1.100 ± 0.004	1.103 ± 0.011
ChCl:Ox (1:1)	1.229 ± 0.003	1.210 ± 0.001
Hydrophobic DES		
Aliq:L Met (3:7)	0.886 ± 0.003	0.887 ± 0.002
Lid:Ac. Dec (1:2)	0.947 ± 0.002	0.949 ± 0.001

**Table 8 molecules-29-03089-t008:** Viscosity (η) values of the DESs prepared using the studied methods at different temperatures.

DES	Viscosity (η) (20 °C)	Viscosity (η) (30 °C)	Viscosity (η) (40 °C)	Viscosity (η) (50 °C)	Viscosity (η) (60 °C)	Viscosity (η) (70 °C)
ChCl:U (1:2)	1697.50 ± 0.05	559.25 ± 0.04	316.13 ± 0.03	97.75 ± 0.02	37.28 ± 0.02	15.76 ± 0.02
ChCl:U (1:2) US	94.62 ± 0.01	55.46 ± 0.01	22.20 ± 0.02	13.46 ± 0.01	7.16 ± 0.01	5.16 ± 0.03
ChCl:EG (1:2)	16.46 ± 0.02	10.30 ± 0.01	8.94 ± 0.04	6.71 ± 0.02	4.81 ± 0.01	4.03 ± 0.02
ChCl:EG (1:2) US	15.22 ± 0.02	10.19 ± 0.03	8.11 ± 0.01	5.80 ± 0.01	4.28 ± 0.01	4.02 ± 0.04
ChCl:EG (1:3)	10.21 ± 0.02	7.06 ± 0.02	5.13 ± 0.03	4.08 ± 0.02	3.59 ± 0.01	2.68 ± 0.02
ChCl:EG (1:3) US	10.10 ± 0.01	6.77 ± 0.02	5.50 ± 0.03	4.51 ± 0.02	3.46 ± 0.01	3.03 ± 0.01
ChCl:EG (1:4)	11.21 ± 0.01	7.33 ± 0.05	5.0 ± 0.03	3.70 ± 0.02	3.18 ± 0.02	2.52 ± 0.02
ChCl:EG (1:4) US	8.38 ± 0.02	6.72 ± 0.02	4.74 ± 0.02	3.77 ± 0.01	3.14 ± 0.01	2.75 ± 0.01
ChCl:Ox (1:1)	1746.75 ± 0.04	491.58 ± 0.03	272.89 ± 0.05	103.36 ± 0.03	35.49 ± 0.02	17.14 ± 0.06
ChCl:Ox (1:1) US	267.53 ± 0.01	126.37 ± 0.04	50.60 ± 0.03	16.53 ± 0.02	7.85 ± 0.02	4.83 ± 0.01
Ali:L Met (3:7)	608.90 ± 0.03	224.34 ± 0.04	75.40 ± 0.01	27.39 ± 0.01	12.34 ± 0.01	7.64 ± 0.06
Ali:L Met (3:7) US	875.36 ± 0.04	284.52 ± 0.05	121.50 ± 0.01	52.75 ± 0.01	21.20 ± 0.01	12.58 ± 0.01
Lid:Ac. Dec (1:2)	346.54 ± 0.03	114.52 ± 0.03	40.51 ± 0.01	15.93 ± 0.01	7.37 ± 0.01	4.81 ± 0.05
Lid:Ac. Dec (1:2) US	398.15 ± 0.06	160.86 ± 0.04	68.27 ± 0.03	27.70 ± 0.02	9.73 ± 0.01	4.90 ± 0.02

**Table 9 molecules-29-03089-t009:** Water content in the prepared DESs.

DES	Water Content (g/L) Heating–Stirring	Water Content (g/L) Ultrasound
Hydrophilic DES		
ChCl:U (1:2)	25.43 ± 1.04	38.92 ± 2.05
ChCl:EG (1:2)	24.09 ± 0.77	24.15 ± 1.02
ChCl:EG (1:3)	20.17 ± 0.80	20.21 ± 0.64
ChCl:EG (1:4)	22.90 ± 1.65	23.16 ± 2.42
ChCl:Ox (1:1)	45.97 ± 4.12	55.72 ± 10.40
Hydrophobic DES		
Aliq:L Met (3:7)	15.44 ± 1.61	4.07 ± 0.6
Lid:Ac. Dec (1:2)	3.92 ± 0.61	3.71 ± 0.37

**Table 10 molecules-29-03089-t010:** Activation energy values obtained for each DES and values of η_0_.

Hydrophilic DES	E_η_ (KJ/mol)	η_0_
ChCl:U (1:2)	77.90	2.37 × 10^−11^
ChCl:U (1:2) US	50.75	8.58 × 10^−8^
ChCl:EG (1:2)	22.77	1.33 × 10^−3^
ChCl:EG (1:2) US	23.02	1.26 × 10^−3^
ChCl:EG (1:3)	21.45	1.45 × 10^−3^
ChCl:EG (1:3) US	19.75	2.85 × 10^−3^
ChCl:EG (1:4)	24.57	4.30 × 10^−4^
ChCl:EG (1:4) US	19.39	2.91 × 10^−3^
ChCl:Ox (1:1)	76.35	4.14 × 10^−11^
ChCl:Ox (1:1) US	70.62	1.48 × 10^−10^
**Hydrophobic** **DES**		
Ali:L Met (3:7)	75.76	1.81 × 10^−11^
Aliq:L Met (3:7) US	71.44	1.48 × 10^−10^
Lid:Ac. Dec (1:2)	73.32	2.60 × 10^−11^
Lid:Ac. Dec (1:2) US	74.70	7.45 × 10^−12^

**Table 11 molecules-29-03089-t011:** Chemical reagents used for the synthesis of DESs.

Chemical Reagent	Reagent Name	Manufacturer	Mass Fraction Purity
C_5_H_14_NOCl	Choline chloride	PanReac AppliChem Castellar del Vallès, Spain	High purity grade
CO(NH₂)₂	Urea	PanReac AppliChem Castellar del Vallès, Spain	>99% (N)
CH_2_OH-CH_2_OH	Ethylene glycol	PanReac AppliChem Castellar del Vallès, Spain	Pure
H_2_C_2_O_4_.2H_2_O	Oxalic acid 2-hidrate	PanReac AppliChem Castellar del Vallès, Spain	99%
C_10_H_20_O	L-Menthol	Thermo Fisher Scientific, Waltham, MA, USA	99%
[CH_3_(CH_2_)_7_]_3_NCH_3_Cl	Aliquat 336 TG	Thermo Fisher Scientific Waltham, MA, USA	-
C_14_H_22_N_2_O	Lidocaine	Thermo Fisher Scientific Waltham, MA, USA	97.5%
CH₃(CH₂)₈COOH	Decanoic acid	Thermo Fisher Scientific Waltham, MA, USA	99%

**Table 12 molecules-29-03089-t012:** Ratios for the synthesis of hydrophilic and hydrophobic DESs.

DES	HBA	HBD	HBA:HBD Ratio
ChCl:U (1:2)	Choline chloride	Urea	1:2
ChCl:EG (1:2)	Choline chloride	Ethylene glycol	1:2
ChCl:EG (1:3)	Choline chloride	Ethylene glycol	1:3
ChCl:EG (1:4)	Choline chloride	Ethylene glycol	1:4
ChCl:Ox (1:1)	Choline chloride	Oxalic acid	1:1
Aliq:L Met (3:7)	Aliquat 336	L-Menthol	3:7
Lid:Ac. Dec (1:2)	Lidocaine	Decanoic acid	1:2

## Data Availability

Data are contained within the article and Appendix A.

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
