# Peer review of "Synthesis and Properties of Hydrophilic and Hydrophobic Deep Eutectic Solvents via Heating-Stirring and Ultrasound"

_molecules, 2024, doi:10.3390/molecules29133089_

Round 1

Reviewer 1 Report

Comments and Suggestions for Authors

The manuscript deals with the comparison of Deep Eutectic Solvents prepared by conventional heating method and by ultrasound. I think that the manuscript Is well written, and the results well reported. I suggest publication after major revisions.

My comments are reported below:

- the introduction (line 40-41) reports a vague definition of DESs. They can be defined more precisely with the physical chemical definition. See the paper:

https://doi.org/10.1007/s10953-018-0793-1

- The method of the microwave antenna put directly into the reaction mixture can be added to the introduction (in the discussion of various synthetic methods used to prepare DESs). This method is proposed as an improvement of the traditional microwave oven. It was also used to prepare choline-based DES. See the following papers:

https://doi.org/10.1016/j.molliq.2022.121104

- https://doi.org/10.1016/j.crgsc.2022.100337

-In all the FTIR spectra: the spectrum of ChCl seems shifted respect to the others, e.g. in Figure 2 the peak at 3320 cm-1 of ChCl is more on the right than the peak at 3316 of the DES.

-Table 6: check the significant figures of the error: a value of 1.0953±0.0021 is not formally right.

-Concerning the viscosity measurements: when you explain the hole theory, make sure to explain that it is just a theory and that it may not work for all the systems. Besides, it is not clear why you have more water in the DESs prepared with ultrasounds if you use a water bath. Do you have control on the water present in your DESs or not? If not, the all manuscript loose his sense. If you have it, you need to quantify the quantity of water that there is in one case or in the other, because it affects many important properties of the DES. See the papers:

-10.12693/APhysPolA.130.239

-https://doi.org/10.1016/j.foodchem.2015.03.123

-10.1039/c8cs00325d

-10.1038/srep29225

-Line 403: there is a typo.

- Moreover, the viscosity measurements need a standard deviation and an error to determine if there is a difference among the different samples or not. Also you need to repeat the measurements and to report an the error for the calculation of activation energy of viscosity. The value of R2 of 0.96 is in reality not a good value for this kind of systems.

-Besides, you use a viscosimeter and not a rheometer to calculate viscosity, therefore you admit that they have a Newtonian behaviour. This may be used as assumption but it needs to be specified. See the paper for further information:

-https://doi.org/10.1016/j.molliq.2019.01.002

- Don’t you use any pretreatment to remove the initial water present in ChCl prior to the synthesis? Do you know how much is the initial water? It may affect the final DES properties as ChCl is highly hygroscopic.

- Finally: all the properties reported are ok, but the main property, which define by definition DES is the melting point. Under this perspective, I believe that your work has sense if you add a comparison of the melting points of the samples prepared with different methods.  

Author Response

List of changes or a rebuttal against each point

Manuscript ID: molecules-3015289
Title: Synthesis and properties of hydrophilic and hydrophobic Deep Eutectic
Solvents by different methods.

Reviewers' comments:

REVISOR #1

The manuscript deals with the comparison of Deep Eutectic Solvents prepared by conventional heating method and by ultrasound. I think that the manuscript Is well written, and the results well reported. I suggest publication after major revisions.

My comments are reported below:

 - the introduction (line 40-41) reports a vague definition of DESs. They can be defined more precisely with the physical chemical definition. See the paper:

https://doi.org/10.1007/s10953-018-0793-1

The authors have modified the introduction (line 40-41) considering the aspect indicated by the reviewer. They have consulted the indicated paper to make the modification. The authors have added this work to the references.

- The method of the microwave antenna put directly into the reaction mixture can be added to the introduction (in the discussion of various synthetic methods used to prepare DESs). This method is proposed as an improvement of the traditional microwave oven. It was also used to prepare choline-based DES. See the following papers:

- https://doi.org/10.1016/j.molliq.2022.121104

- https://doi.org/10.1016/j.crgsc.2022.100337

The authors have added the method of the microwave antenna to the introduction. They have consulted the indicated papers. The authors have added these two works to the references.

 -In all the FTIR spectra: the spectrum of ChCl seems shifted respect to the others, e.g. in Figure 2 the peak at 3320 cm-1 of ChCl is more on the right than the peak at 3316 of the DES.

3320 cm-1 is a wrong value. The actual value is 3220 cm-1. This value has been corrected in Figures 1, 2 and 3 and in the text of the work.

-Table 6: check the significant figures of the error: a value of 1.0953±0.0021 is not formally right.

In Table 6, the significant figures of the error have been checked and revised.

 -Concerning the viscosity measurements: when you explain the hole theory, make sure to explain that it is just a theory and that it may not work for all the systems. Besides, it is not clear why you have more water in the DESs prepared with ultrasounds if you use a water bath. Do you have control on the water present in your DESs or not? If not, the all manuscript loose his sense. If you have it, you need to quantify the quantity of water that there is in one case or in the other, because it affects many important properties of the DES. See the papers:

-10.12693/APhysPolA.130.239

-https://doi.org/10.1016/j.foodchem.2015.03.123

-10.1039/c8cs00325d

-10.1038/srep29225

Taking into account the reviewer's suggestion, the authors have explained that Hole's theory is a theory that can work in some systems and not in others. They have also measured the water content in the DESs prepared by the two methods (see Table 10). The authors have added one of the papers (10.1039/c8cs00325d) to the references.

Thanks for your suggestions.

Line 403: there is a typo.

This typo have been corrected.

- Moreover, the viscosity measurements need a standard deviation and an error to determine if there is a difference among the different samples or not. Also you need to repeat the measurements and to report an the error for the calculation of activation energy of viscosity. The value of R2 of 0.96 is in reality not a good value for this kind of systems.

This point has been modified according to the suggestion of the review. Figure 8 has been eliminated and replaced by Table 8 (new viscosity measurements). The viscosity measurements have been repeated to calculate the activation energy (Figure 10).

-Besides, you use a viscosimeter and not a rheometer to calculate viscosity, therefore you admit that they have a Newtonian behavior. This may be used as assumption but it needs to be specified. See the paper for further information:

-https://doi.org/10.1016/j.molliq.2019.01.002

The authors have modified this point considering the aspect indicated by the reviewer. They have added this work to the references.

 - Don’t you use any pretreatment to remove the initial water present in ChCl prior to the synthesis? Do you know how much is the initial water? It may affect the final DES properties as ChCl is highly hygroscopic.

The authors didn´t use any pretreatment to remove the initial water in ChCl prior to the synthesis.

They didn´t do pretreatment to remove the moisture because when opening the bottle of reagent, they wasn´t observed moisture.  Later the amount necessary for prepare the DESs was placed in a desiccator so that it did not collect humidity. But it is possible that ChCl, due to its high hygroscopicity, had some initial moisture, which is why choline chloride DESs have been confirmed to contain more water.

- Finally: all the properties reported are ok, but the main property, which define by definition DES is the melting point. Under this perspective, I believe that your work has sense if you add a comparison of the melting points of the samples prepared with different methods. 

The authors, following the reviewer's suggestions, have carried out the thermal analysis study of the synthesized DESs (differential scanning calorimetry) to know the melting points of the DESs. The DSC curves have been added and commented in the text of the work.

Reviewer 2 Report

Comments and Suggestions for Authors

The authors aimed to synthesize both hydrophobic and hydrophilic DES using ultrasound as a preparation method. However, the depth of exploration into recent advances in this field within the main document is limited and could be expanded to enhance the publication's quality.

The definition of DES in the introduction needs to be revised, as a DES by definition need to present deviations from the solid-liquid phase behavior.

To emphasize that this method minimizes the degradation of initial compounds, would be ideal to present novelty. This can particularly be shown with DESs containing oxalic acid, which need to be evaluated. The question of whether esterification also occurs and to what extent the ultrasound method can prevent it needs to be addressed.

Chemical shifts and bibliographic densities can be provided in supplementary information.

The reduction in viscosity demonstrated by this method is one of the most significant results, since it facilitates the application of DES when aqueous solutions are not envisaged. However, since ChCl is a highly hygroscopic compound, changes in the degree of hydration may be the reason for the observed differences, as the heating method also causes the evaporation of some residual water. To prove the viscosity differences, ChCl should be dried before the preparation of the DES and then transferred to sealed flasks.
More specific details should be provided in the experimental section.

Comments on the Quality of English Language

The abstract and introduction sections need specific attention and revision. Therefore, it is advisable to strive for a more fluid and cohesive text that helps readers following the progression of ideas, understanding the main goal of the work, and identifying the main observed advantages. Additionally, careful reading of the document is also advised, as typographical errors such as lack of punctuation, unusual symbols like "+" at the beginning of the text, etc., may be present throughout. 

Author Response

List of changes or a rebuttal against each point

Manuscript ID: molecules-3015289
Title: Synthesis and properties of hydrophilic and hydrophobic Deep Eutectic
Solvents by different methods.

Reviewers' comments:

REVISOR #2

The authors aimed to synthesize both hydrophobic and hydrophilic DES using ultrasound as a preparation method. However, the depth of exploration into recent advances in this field within the main document is limited and could be expanded to enhance the publication's quality.

The definition of DES in the introduction needs to be revised, as a DES by definition need to present deviations from the solid-liquid phase behavior.

The introduction have been revised considering the aspects indicated by the reviewer about the definition of DES.

To emphasize that this method minimizes the degradation of initial compounds, would be ideal to present novelty. This can particularly be shown with DESs containing oxalic acid, which need to be evaluated. The question of whether esterification also occurs and to what extent the ultrasound method can prevent it needs to be addressed.

The authors thank the reviewer for the suggestion to reevaluate the DESs containing oxalic acid. The new studies have been incorporated into the work in the form of a commentary on Figures 6 and 7 and Table 4.

Chemical shifts and bibliographic densities can be provided in supplementary information.

 Bibliographic densities have been provided in supplementary information (Table S1).

The reduction in viscosity demonstrated by this method is one of the most significant results, since it facilitates the application of DES when aqueous solutions are not envisaged. However, since ChCl is a highly hygroscopic compound, changes in the degree of hydration may be the reason for the observed differences, as the heating method also causes the evaporation of some residual water. To prove the viscosity differences, ChCl should be dried before the preparation of the DES and then transferred to sealed flasks.
More specific details should be provided in the experimental section.

The authors didn´t use any pretreatment to dried the ChCl prior to the synthesis. They didn´t do pretreatment because when opening the bottle of reagent, they wasn´t observed moisture.  Later the amount necessary for prepare the DESs was placed in a desiccator so that it did not collect moisture. The prepared DESs were transferred to sealed flasks. But it is possible that ChCl, due to its high hygroscopicity, had some initial moisture, which is why choline chloride DESs have been confirmed to contain more water.

This phrase “The prepared DESs were transferred to sealed flasks” has been added in the section 3.2.- Synthesis method.

Comments on the Quality of English Language

The abstract and introduction sections need specific attention and revision. Therefore, it is advisable to strive for a more fluid and cohesive text that helps readers following the progression of ideas, understanding the main goal of the work, and identifying the main observed advantages. Additionally, careful reading of the document is also advised, as typographical errors such as lack of punctuation, unusual symbols like "+" at the beginning of the text, etc., may be present throughout. 

The entire text has been thoroughly revised for English language accuracy. Typos, including missing punctuation and unusual symbols such as "+" at the beginning of the text, have been corrected.

Reviewer 3 Report

Comments and Suggestions for Authors

Although there is a marked interest in improving the moethos for preparation of DES, both in terms of energy efficiency and time consumed, the presented manuscript is more a collection of preparation methods carried out by other authors, and provides few results to prove that the ultrasound assisted method is in fact the best one to prepare DES.

The authors should provide more data in order for the manuscript to be more sound and original. For example, the NMR analysis of the DES prepared by the ultrasound method should be provided, in order to see if the esterification reaction, that occurs between the DES components, is in fact avoided using this preparation method.

In the abstract section, the first sentence describes DES as a green alternative. But a green alternative compared to what? This should be rephrased.

In the introduction part, in Line 100, the authors say that the microwave and ultrasound preparation methods cosume less energy that the heating/stirring, but this is not in accordance with the values presented, since the heating/stirring method consumes less energy that the microwave one. This should be corrected.

In fact, stability studies of the prepared DES, using both methods, should be also presented, and the extent of esterification should also be determined.

The water content of the prepared DES should also be referred. Water amount in DES is crucial in order to study and compare its phisicochemical properties, in particular viscosity.This has been reported by several authors, and is referred to in this document, however no information on this is provided. This values would help to justify some of the differences in the presneted values of viscosity, and even density.

Comments on the Quality of English Language

The overall English of the document should be revised, with several gramatical errors being noticed.. Examples are in line 42, where it should read " they do NOT require", or line 50, where "acids carboxilic" should read "carboxilic acids.

The Abstract section should be completelly re-written, since the phrasing is incorrect, and it conatins several errors.

Author Response

List of changes or a rebuttal against each point

Manuscript ID: molecules-3015289
Title: Synthesis and properties of hydrophilic and hydrophobic Deep Eutectic
Solvents by different methods.

Reviewers' comments:

REVISOR #3

Although there is a marked interest in improving the methods for preparation of DES, both in terms of energy efficiency and time consumed, the presented manuscript is more a collection of preparation methods carried out by other authors, and provides few results to prove that the ultrasound assisted method is in fact the best one to prepare DES.

The work has been modified and expanded with different tests and new measures to improve the quality of the research carried out and demonstrate the best characteristics of the DESs prepared by the ultrasound method.

The authors should provide more data in order for the manuscript to be more sound and original. For example, the NMR analysis of the DES prepared by the ultrasound method should be provided, in order to see if the esterification reaction, that occurs between the DES components, is in fact avoided using this preparation method.

The authors thank the reviewer for the suggestion to evaluate the NMR analysis of all the DES prepared by the ultrasound method. The new studies have been incorporated into the text in the form of a commentary on Figures 7 and 9 and Tables 2-5.

In the abstract section, the first sentence describes DES as a green alternative. But a green alternative compared to what? This should be rephrased.

The authors have modified the first sentence as indicated by the reviewer. The modified phrase is:

Deep eutectic solvents (DESs) have emerged as a greener alternative to other more polluting traditional solvents…

In the introduction part, in Line 100, the authors say that the microwave and ultrasound preparation methods consume less energy that the heating/stirring, but this is not in accordance with the values presented, since the heating/stirring method consumes less energy that the microwave one. This should be corrected.

This error has been corrected as indicated by the reviewer.

In fact, stability studies of the prepared DES, using both methods, should be also presented, and the extent of esterification should also be determined.

New studies have been carried out on all the DESs prepared by the two methods (NMR spectra, water present in the DESs  and comparison of melting points). These new tests have been incorporated into the revised document in the form of figures, tables and comments on the extent of esterification.

The water content of the prepared DES should also be referred. Water amount in DES is crucial in order to study and compare its physicochemical properties, in particular viscosity. This has been reported by several authors, and is referred to in this document, however no information on this is provided. This values would help to justify some of the differences in the presented values of viscosity, and even density.

They authors have studied the water content of prepared DESs by method Karl Fisher incluying these results in the work (see Table 10).

Comments on the Quality of English Language

The overall English of the document should be revised, with several grammatical errors being noticed.. Examples are in line 42, where it should read " they do NOT require", or line 50, where "acids carboxylic" should read "carboxylic acids.

The Abstract section should be completely re-written, since the phrasing is incorrect, and it contains several errors.

The revision of the English language of the document has been carried out. Typos (as in line 42, where it should read " they do NOT require", or line 50, where "acids carboxylic" should read "carboxylic acids”, etc.) have been corrected. The Abstract has been modified and re-written.

Round 2

Reviewer 1 Report

Comments and Suggestions for Authors

I thank the author for the hard work performed to improve the manuscript. With the corrections i think that the work gains value and it is suitable for pubblication.

Reviewer 3 Report

Comments and Suggestions for Authors

The authors have adressed the previous comments, and the manuscript is improved.

Comments on the Quality of English Language

The authors have edited the text and improved the English